# Long-Term but Not Short-Term Maternal Fasting Reduces Nephron Number and Alters the Glomerular Filtration Barrier in Rat Offspring

**DOI:** 10.3390/life11040318

**Published:** 2021-04-06

**Authors:** Abdullah Alshamrani, Waleed Aldahmash, Fawaz Falodah, Maria Arafah, Abdel Halim Harrath, Saleh Alwasel

**Affiliations:** 1Zoology Department, College of Science, King Saud University, Riyadh 11451, Saudi Arabia; 437106982@student.ksu.edu.sa (A.A.); Waldahmash@ksu.edu.sa (W.A.); ffalodah@ksu.edu.sa (F.F.); hharrath@ksu.edu.sa (A.H.H.); 2Department of Pathology, College of Medicine, King Saud University, Riyadh 11451, Saudi Arabia; mariaarafah@ksu.edu.sa

**Keywords:** fasting, fetal programming, kidney, barrier

## Abstract

The present study examined the effects of maternal Ramadan-type fasting during selected days in the first, second, or third trimester, or during the entire pregnancy, on the kidney structure of male rat offspring. Pregnant rats were provided with food ad libitum during pregnancy (control group, C), or they were exposed to 16 h of fasting/day for three consecutive days in the middle of the first (FT1), second (FT2), or third trimester (FT3), or during whole pregnancy (FWP). Our results showed that dams in the FWP group demonstrated lower food intake and body weight during gestation. Litter size was unaltered by fasting in all groups; however, litter weight was significantly reduced only in the FWP group. Nephron number was decreased in the FWP group, but it remained unchanged in the other fasting groups. The ultrastructure of the glomerular filtration barrier indicated that the kidneys of offspring of the FWP group demonstrated wider diameters of fenestrations and filtration slits and smaller diameters of basement membranes. This was reflected by a significant increase in proteinuria in FWP only. These results suggest that, unlike with short-term fasting, which seems to be safe, maternal long-term fasting induces structural changes that were non-reversible, and that may contribute to impaired renal function, leading to chronic diseases in later life.

## 1. Introduction

Fetal development is among the most complex biological processes, especially in mammals. During fetal development, specific time periods exist during which particular organ systems develop. Organogenesis occurs in very delicate uterine environmental conditions, and changes in one or more of these conditions may negatively affect the structure of one or more organs. The fetal programming hypothesis suggests that an inadequate uterine environment impairs organogenesis and induces permanent fetal adaptations, resulting in fetal programming of chronic diseases in later stages of life [1]. Fetal undernutrition in middle-to-late gestation has been shown to increase mortality from cardiovascular disease [2]. One of the main maternal factors involved in fetal programming of hypertension is poor diet during pregnancy [3]. Many epidemiological and experimental studies provided clear evidence to support this hypothesis, which has since been renamed the developmental origins of health and disease (DOHaD) [4,5,6,7,8,9,10].

The fetus, when experiencing a shortage in nutritional supply, redirects limited nutritional resources to critical organ systems, such as the brain and the heart, at the expense of other non-functional organs, including the kidneys [11]. Animal models provide an invaluable tool to study the underlying mechanisms of developmental programming. Different maternal poor diets were implemented to induce fetal programming of the kidney in rat offspring, including maternal food restriction [8], 50% protein restriction [10], mineral restriction [11], and a high-fat diet [12]. Rat offspring exposed to maternal food restriction were smaller at birth, exhibited lower nephron numbers, and exhibited higher blood pressure compared to adult control offspring [12,13]. Despite exposure to food restriction in utero reducing nephron number in rat offspring, the total glomerular filtration rate was maintained at normal levels, suggesting that single-glomerular filtration rate could be increased [14]; the ultrastructure of the glomerular filtration barrier in adult rat offspring exposed to maternal food restriction may have been adapted for increased single-glomerular filtration rate, as the fenestrations were wider, the basement membrane was less condensed, and filtration slits were wider [14].

The changes in water and food intake, food type, physical activities, and sleeping patterns due to fasting may disturb maternal metabolism and contribute to fetal programming. Studies investigating the effect of Ramadan fasting on maternal or fetal health are still limited, and they show conflicting results due to their small sample sizes [15]. Some studies found that fasting during pregnancy significantly lowered maternal macronutrient intake and body weight [16,17,18]. In line with reduced gestational weight gain, some studies reported a reduction in baby birth weight [19]. Other studies, however, showed a reduction in placental weight, but not fetal weight. In a previous study, we showed that rat offspring exposed to maternal Ramadan-type fasting during whole pregnancy displayed delayed nephrogenesis and less well-differentiated glomeruli at birth [20]. Nephron number was reduced by 30%, but glomerular size was significantly increased. In Islam, pregnant Saudi women fast only for one-ninth of the gestation period. In addition, their fasting could fall at any frame of the gestation period. An important limitation of our previous study was that offspring were exposed to maternal fasting 16 h per day during the whole gestation period, in order to maximize the effect of fasting. The current study, therefore, aimed to compare the effects of maternal Ramadan-type fasting for three consecutive days during the first, second, or third trimester to fasting during the whole gestation period on renal histology and ultrastructure.

## 2. Materials and Methods

### 2.1. Animal Model

Virgin female Wistar rats (*Rattus norvegicus*) were purchased from the Animal House of the Zoology Department, King Saud University. Rats were housed for one week in standard cages, in a light-controlled room, where the light was on between 8:00 a.m. and 4:00 p.m. Room temperature ranged between 23 and 25 °C. Rats were provided free access to water and standard rat chow (ingredients are shown in Table 1; Saudi Grains Organization). After one week of acclimatization, healthy male (250–300 g) and female (225–250 g) rats were assigned into mating cages. Mating was confirmed by the presence of a sperm plug. Pregnant rats were randomly assigned into five groups. The first, second, and third group underwent 3 days of Ramadan-type fasting (RTF) during the first, second, or third trimester of gestation, respectively (TF1, *n* = 6; TF2, *n* = 6; TF3, *n* = 6). The 3 days of fasting was assumed to be equivalent to one month of fasting in humans. Group number 4 fasted during the whole pregnancy (FWP, *n* = 6). Food and water were removed between 4:00 p.m. and 8:00 a.m. in all RTF groups, initiating a daily fasting of 16 h. The fifth group was used as a control (C, *n* = 6), where pregnant rats were given a free access to food and water from the beginning of gestation until delivery. After birth, all dams were switched to receive free access to food and water. This study was revised and approved by the Ethics Committee at King Saud University (KSU-SE-19-69).

### 2.2. General Measurements

Rats were monitored every 12 h to determine the presence of the sperm plug and note delivery time, so that gestational length could be calculated to the nearest 12 h. Female body weight was recorded before mating, every day during gestation, and after delivery. Maternal food and water intake were recorded daily until birth. The ratio of maternal body weight at delivery to body weight before pregnancy was calculated to examine the contribution of maternal nutrient storage to maintaining fetal weight. Gestational weight was calculated by subtracting initial body weight from maternal body weight before delivery. Litter size, weight, and male-to-female ratio were recorded. Average birth weight was calculated for both sexes. Next, the body weight of 4-week-old male offspring was recorded before dissection; kidneys were weighed to the nearest milligram, and kidney weight as a percentage of body weight was calculated. Then, heart weight was used as a reference organ.

### 2.3. Light Microscopy

The kidneys of 4-week-old male offspring (one from each litter) were harvested for examination under a light microscope. Kidneys were fixed for 24 h in 10% neutral buffered formalin. Samples were dehydrated by using serial concentrations of ethanol, followed by clearing with xylene and embedding in paraffin wax. Once in paraffin blocks, the right kidney was cut into 5 µm–thick sections for general histological examination. All sections were deparaffinized and stained with hematoxylin and eosin (H&E). Renal histology was examined and photographed using a Nikon Eclipse 80i microscope with a Nikon DXM 1200C digital camera. Glomerular size was estimated in 10 fields in 10 sections per kidney, using IM500 Image Manage software.

### 2.4. Glomerular Number

Glomerular number was estimated by using a modified version of the physical fractionator technique. Briefly, the left kidney was cut into serial 20 µm–thick sections for estimation of glomerular number. Using intervals of 15 sections, 6 section pairs were used as reference and look-up sections. Sections were stained with H&E, and the first section pair was sampled using a random-number table. Starting with the reference section, glomeruli were counted in a frame, and then they were compared to the corresponding part of the look-up section. Glomeruli that were present on both the reference section and the look-up section were discounted. The counting frame was moved to the next field, and the process was repeated in a zigzag shape until the whole section was counted. The following formula was used to calculate the total number of glomeruli:$$\mathrm{Total}\;\mathrm{number}\;\mathrm{of}\;\mathrm{glomeruli}=\left(\;\sum Q\;\times \;\frac{1}{n}\;\right)\;\times \;2$$
where Q indicates the number of glomeruli counted, and n indicates the fraction of section pairs counted. Since glomeruli were counted bi-directionally (reference section versus look-up section and vice versa), the outcome was multiplied by 2.

### 2.5. Electron Microscopy

Renal cortex tissue was dissected from 28-day-old offspring and fixed in Karnovsky’s fixative (pH 7.2) at 4 °C for 12 h. Tissue was washed in cacodylate buffer and post-fixed in 2% osmium tetroxide for 1 h. Then, tissue was processed in Agar 100 proxy resin. Semi-thin sections (approximately 1 µm) were cut and stained with toluidine blue. The block was trimmed toward the outer cortical glomeruli where possible. Ultra-thin sections (70–90 nm) were obtained by using an ultramicrotome (Leica EM UC7, Wetzlar, Germany) and were double-stained with uranyl acetate and lead citrate. Photomicrographs were taken using a JEOL 100 CX electron microscope. Compartments of the glomerular filtration barrier were examined, and the diameters of glomerular endothelial cell fenestrations, the basement membrane, and filtration slits were measured at 32 random spots in each photomicrograph, at a magnification of 25,000×, using image-manager software (Leica IM 1500).

### 2.6. Urine Albumin Analysis

Urine was collected from 4-month-old offspring, using individual metabolic cages. Urinary albumin concentration was evaluated using Nephrat II Rat Albumin ELISA kit (Ethos Biosciences).

### 2.7. Apoptosis

Apoptosis was estimated by using a TUNEL assay. Paraffin sections were immersed in xylene and then rehydrated in a descending series of alcohol concentrations. Sections were washed in distilled water, immersed in phosphate-buffered saline, and then incubated with proteinase K for 15 min. Tissue sections were permeabilized using 0.1% Triton X-100 with sodium citrate. This was followed by addition of 0.3% pepsin in HCl (pH 2) for 5 min, at 37 °C. Sections were submerged in citrate buffer and placed in a microwave at 750 W for 45 s, followed by two washing steps with phosphate-buffered saline. The In Situ Cell Death Detection kit (TMR-red, Roche Diagnostics, Mannheim, Germany) was used for TUNEL staining, according to the manufacturer’s instructions. A positive control sample was treated with recombinant DNase-I for 10 min, at room temperature, in order to induce DNA fragmentation. Some slides were treated without Terminal deoxynucleotidyl transferase (TdT) as negative controls. All sections were stained with Hoechst dye, washed in TE buffer, and mounted in 50% glycerol/TE. Sections were photographed using a Nikon TE 2000 fluorescence microscope connected to a Nikon DS-cooled camera.

### 2.8. Statistical Analyses

Results are expressed as means ± standard deviation. Differences between groups were determined by using one-way ANOVA. Differences were considered significant at *p* < 0.05. All statistical analyses were performed by using the SPSS program (Version 20).

## 3. Results

This section may be divided by subheadings. It should provide a concise and precise description of the experimental results, their interpretation, and the experimental conclusions that can be drawn.

### 3.1. Maternal Water and Food Intake

Table 2 shows maternal and litter measurements during gestation and after birth. Maternal body weight at mating was similar in all groups; however, the body weight of dams in the FT3 and FWP groups was significantly lower than other groups at the last day of gestation. This led to a significant reduction in gestational weight gain in the FT3 and FWP groups, by 21% and 34%, respectively, in comparison to the control group.

Total maternal food intake during gestation was only reduced in the FWP group (Table 2); however, as shown in (Figure 1), food intake was significantly reduced during fasting days of the FT1, FT2, and FT3 groups, in comparison to the control group. Figure 2 shows the daily body weight of pregnant rats in all groups. In agreement with food intake, the body weight of dams from the FT1, FT2, and FT3 groups decreased significantly on fasting days; however, they were able to catch up rapidly on the following days. Gestational length was reduced only in the FT3 group, as shown in Table 2. Litter size was unaffected, but litter weight was significantly reduced by 28% in the FWP group only. Maternal body weight at delivery was significantly reduced in all fasted groups, compared to the control group; however, in the FWP group, maternal body weight was even smaller than in the other fasting groups (Table 2).

### 3.2. Histological Study

The estimated nephron number in 4-week-old rat offspring exposed to maternal fasting during whole pregnancy was significantly smaller than the nephron number in control offspring, though offspring from the FT1, FT2, or FT3 groups exhibited similar numbers of nephrons as the control offspring (Figure 3). The relative reduction of nephron number in the FWP group was 22%. No significant differences were found in glomerular size between the FT1, FT2, FT3, or FWP and control groups (Figure 4).

### 3.3. Ultrastructure of Glomerular Filtration Barrier

Figure 5 shows the ultrastructure of the glomerular filtration barrier in a representative glomerulus. The diameter of the fenestrations in endothelial cells surrounding glomerular capillaries was increased in rat offspring exposed to maternal FWP. The average diameter of fenestrations in the FT1, FT2, and FT3 groups was comparable to that of the control group (Figure 6A). Similarly, the basement membrane was thinner in the FWP group compared to other groups (Figure 6B). Researchers also noticed that the density of the basement membrane was reduced in the FWP group. The average diameter of the slits was increased in the FWP group, compared to the control group, but it remained unaltered in the FT1, FT2, or FT3 groups (Figure 6C).

### 3.4. Evaluation of Urinary Albumin

Urinary albumin concentration in rat offspring exposed to maternal fasting for whole pregnancy period was significantly higher than control offspring. Offspring from other fasting groups have comparable levels of urinary albumin to the control group (Figure 7).

### 3.5. DNA Cleavage in Kidney Cells

Some kidney cells are programmed to undergo cell death; therefore, some intense staining was detected even in the control group. As shown in Figure 8, Figure 9 and Figure 10, offspring from the FT1, FT2, and FT3 groups exhibited a similar rate of apoptosis as the control group. However, TUNEL assay staining in the kidneys of rats that were exposed to maternal FWP was significantly higher than the control group.

## 4. Discussion

The literature provides clear evidence that maternal malnutrition reduces fetal growth and induces developmental programming [3]. Recently, Mohany et al. [20] showed that maternal fasting for 16 h per day for the entire period of gestation induced critical renal histopathology in rat offspring. In general, fetal kidneys from the fasted group were less developed and exhibited significantly delayed nephrogenesis, as compared to control fetuses at gestational days 18–20. In addition, glomerular number was decreased by 30% in offspring exposed to maternal fasting, suggesting that these rats may develop hypertension. The experimental model used in that study was set to mimic Ramadan fasting in Saudi Arabia, where pregnant women abstain from eating and drinking for 16 h per day during the summer month of Ramadan. Pregnant Saudi women may fast for up to one-ninth of the gestation period [20]; however, in that study, pregnant rats were made to undergo fasting for the whole gestation period, in order to maximize the effects for easy detection. The current study, therefore, was designed to examine the effects of fasting for one-ninth of gestation on the kidneys of the offspring.

Results from the current study show significant differences between fasting for one-ninth of a pregnancy and fasting for the whole pregnancy. Despite rats from fasting groups receiving free access to food for 8 h during fasting days, maternal food intake was significantly reduced in all experimental groups. Conversely, fasting for one-ninth of the pregnancy did not alter the total gestational food intake. The reduction in food intake during pregnancy days (days 6–7, 13–14, or 20–22) led to a significant reduction in maternal body weight in the FT1, FT2, and FT3 groups, respectively. Gestational weight gain was restored in FT1 and FT2, but not FT3, likely because FT3 dams were exposed to fasting at the end of pregnancy and thus were unable to catch up. Interestingly, control dams increased body weight at delivery day by 9%, compared to their initial body weight, perhaps to prepare for weaning. Dams from the FT1, FT2, and FT3 groups were not able to build up body weight, although they were able to maintain their initial body weight. In contrast, dams from the FWP group demonstrated significant reduction in maternal food and water intake at daily and total measurements. The reduction in food intake observed in the FWP group was similar to mild food restriction used by other studies [21,22]. Maternal body weight of rats of the FWP group was significantly reduced in terms of daily measurement and gestational weight gain. Interestingly, FWP dams lost 9% of their initial body weight during pregnancy. This suggests that FWP dams partly use their own body nutrient storage in order to maintain a normal supply of blood nutrients to aid their developing fetuses.

Low birth weight has been used as a biological marker of developmental programming of chronic diseases. The associations between birth weight and disease have been confirmed in a large number of cohort studies worldwide [3]. Epidemiological studies showed that poor growth in fetal life is associated with coronary heart disease [23], diabetes [24], hypertension [25], and obesity [26]. Litter size was not affected by fasting, either for part or the whole of the pregnancy period; however, average pup birth weight was significantly reduced by approximately 14% in the FT1, FT2, and FT3 groups, and by 20% in the FWP group. The reduction in birth weight in the FT1, FT2, and FT3 groups was mild, and it was not associated with any of renal parameters investigated in this study.

A reasonable number of studies in humans and laboratory animals currently exist showing that maternal malnutrition leads to hypertension, mainly by programming the offspring kidney [10,27]. Impairment in nephrogenesis in humans and animals exposed to undernutrition in utero points to the kidney as a targeted organ. A significant reduction in nephron number has been reported in offspring exposed in utero to protein or global nutrient restriction, uterine artery ligation, hyperglycemia, and after exposure to various agents, such as glucocorticoids or alcohol [28,29]. Similar to our previous findings [20], glomerular number was decreased by 22% in offspring exposed to maternal fasting for the whole pregnancy period. Interestingly, the glomerular number was unaffected when dams were exposed to fasting for only one-ninth of the gestation period, regardless of trimester. The reduction in nephron number observed in offspring of the FWP group suggests that maternal fasting for long-term, but not short-term, may slow or halt nephrogenesis. Various mechanisms have been suggested to explain the reduction in nephron number, including a depletion of stem cells, increased apoptosis, inhibition of ureteric branching, or early cessation of nephrogenesis [20,30,31,32].

Few studies assessed the ultrastructure of the glomerular filtration barrier in developmental programming models. One study examined the ultrastructure of glomerular filtration barrier at embryonic day 20 and at 6 months of age for rats exposed to maternal food restriction [14]. This study found a remarkable retardation in the development of the glomerular filtration barrier of fetuses exposed to maternal food restriction. The slits between podocyte foot processes were not fully formed at embryonic day 20, but they were significantly wider at 6 months of age. It also showed that the basement membrane was thicker, but less condensed, at 6 months of age. In the current study, results showed that maternal fasting for whole gestation period, but not for one-ninth of the gestation period, altered the ultrastructure of the glomerular filtration barrier. The diameters of both endothelial cell fenestrations and slits were increased, and in addition, the thickness and density of the basement membrane was reduced. We suspect that these changes may increase the single-glomerular filtration rate in order to counteract the reduction in glomerular number.

Renal function may be decline by offspring exposed to maternal fasting for the whole pregnancy as indicated by the significant increase in urinary albumin concentration compared to the control and other fasting groups. This is consistent with finding from another fetal programming model where exposure to maternal low protein diet led to a significant increase in urinary albumin concentration [33].

Apoptotic cells were significantly increased in the FWP group, but they remained unchanged in other fasting groups, compared to the control group. Apoptosis has been reported in proximal and distal tubules, as well as in interstitial and endothelial cells, with no involvement of glomerular cells [34]. The present study represents the first to provide clear evidence for the occurrence of apoptosis mainly in glomerular cells under fasting. This may explain the decreased number of nephrons in the FWP group, which remained unchanged in other fasting groups. This finding is in agreement with a previous study that showed that apoptosis is the major cell clearance mechanism in glomeruli, counterbalancing the negative impact of many other substances or cell mechanisms [35]. Moreover, the wider diameter of fenestrations and filtration slits of the glomerular filtration barrier in the FWP offspring kidney, as well as the smaller diameter of basement membrane, may have occurred through an apoptotic mechanism. Indeed, previous studies demonstrated that, in many kidney diseases, apoptosis was an ongoing process [35,36].

## 5. Conclusions

In conclusion, our findings suggest that exposure to RTF in utero for one-ninth of the gestation period, regardless of trimester, did not alter nephron number or the ultrastructure of the glomerular filtration barrier in male rat offspring. However, if maternal fasting occurred during the whole gestation, reduced nephron number, glomerular membrane thickness, and increased diameter of both endothelial cell fenestrations and slits may result.

## Figures and Tables

**Figure 1 life-11-00318-f001:**
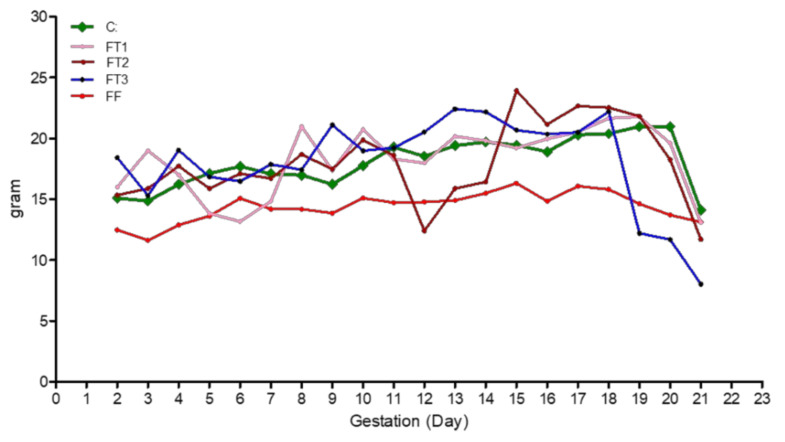
Daily food consumption of dams fasted during the first (FT1), second (FT2), or third trimester (FT3), or the whole pregnancy (FWP).

**Figure 2 life-11-00318-f002:**
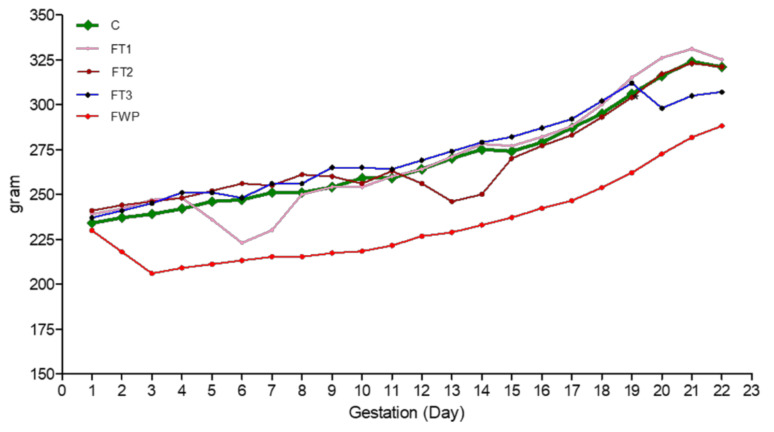
Daily maternal weight of dams during pregnancy until birth.

**Figure 3 life-11-00318-f003:**
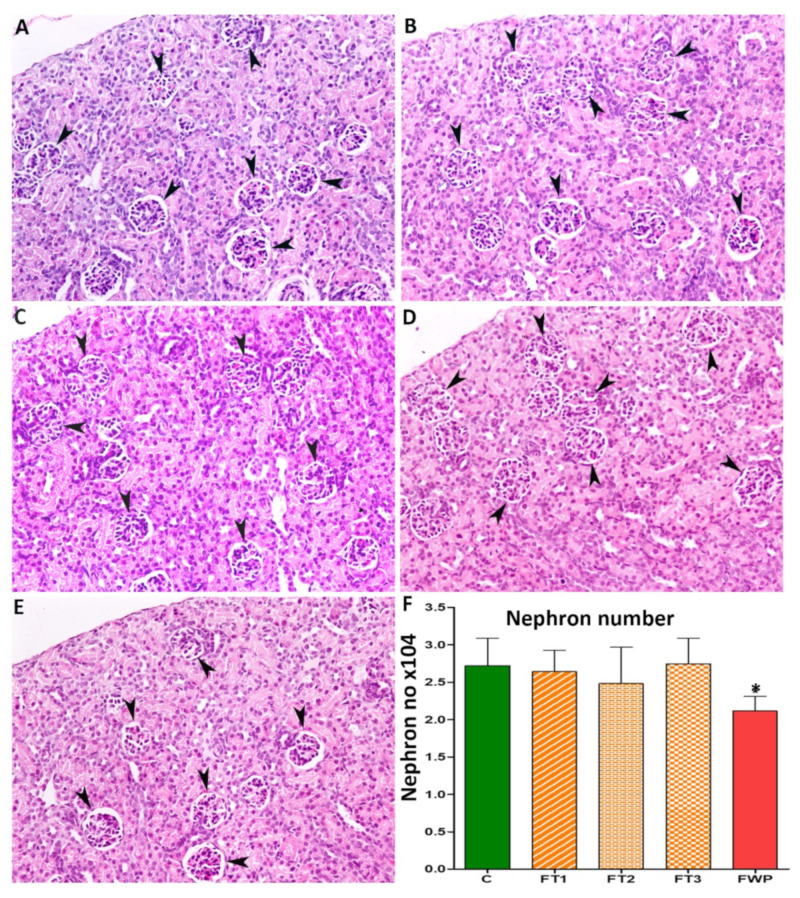
Micrographs and a diagram show the nephron number in (**A**) control rats, (**B**–**D**) rats exposed to fasting in first, second, or third trimester (FT1, FT2, or FT3, respectively), or (**E**) exposed to fasting during whole gestation (FWP). Magnification: 200×; arrow heads indicate glomeruli. (**F**) Estimated nephron number where asterisk (*) is statistically significant.

**Figure 4 life-11-00318-f004:**
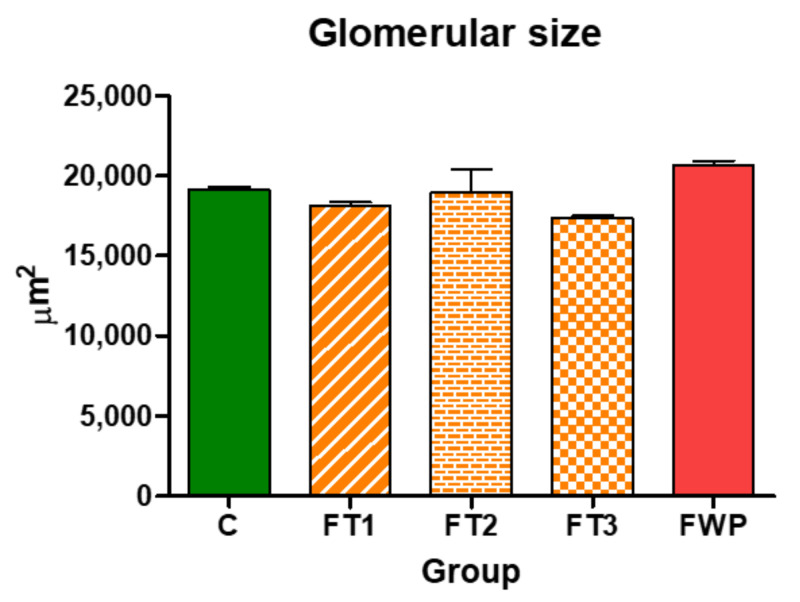
Diagram of glomerular size in all experimental groups.

**Figure 5 life-11-00318-f005:**
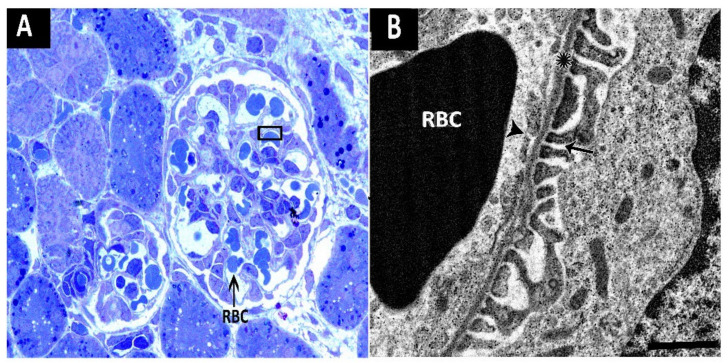
(**A**) A semithin section of a representative glomerulus where the black triangle was magnified in (**B**) micrograph. (**B**) A micrograph showing the ultrastructure of the glomerular filtration barrier at a magnification of 5000×. Asterisk (*) indicates basement membrane; arrows indicate fenestrations; arrow heads indicate slits; RBC, red blood cell.

**Figure 6 life-11-00318-f006:**
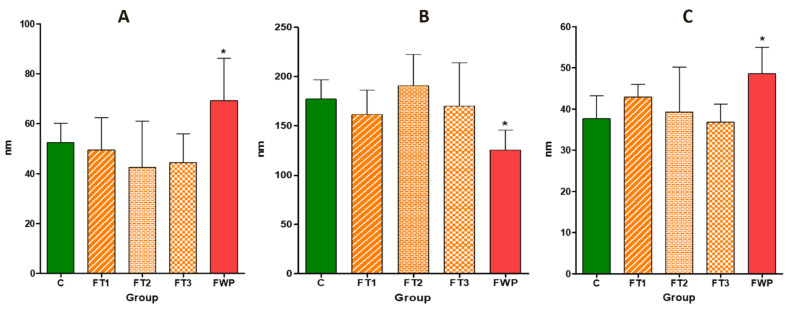
(**A**) The diameter of glomerular endothelial cell fenestrations in the kidneys of rat offspring exposed to maternal fasting in the first (FT1), second (FT2), or third trimester (FT3), or during the whole pregnancy (FWP), compared to the control group, C. (**B**) The thickness of the glomerular filtration basement membrane in all experimental groups. (**C**) The thickness of filtration slits in all experimental groups. Asterisk (*) indicates that *p* is less than 0.05.

**Figure 7 life-11-00318-f007:**
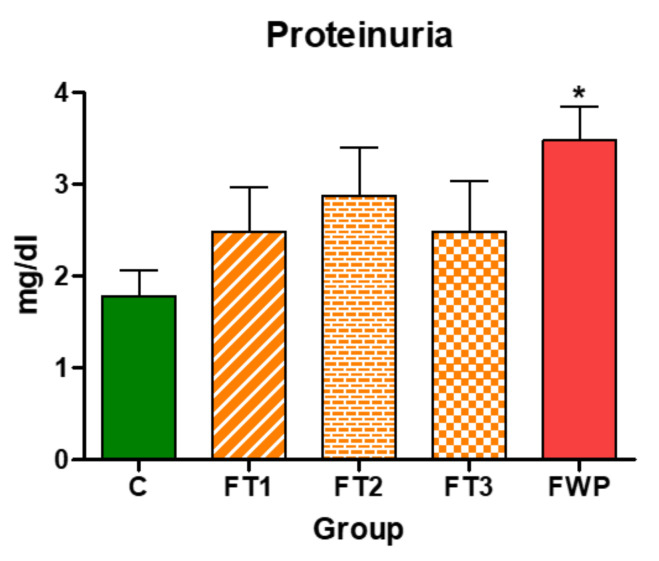
Urinary albumin concentration in 4-month-old offspring. Albumin concentration in urine from rat offspring exposed to maternal fasting during whole pregnancy (FWP) was significantly higher than in control (C) or offspring exposed to maternal fasting during first trimester (FT1), second trimester (FT2), or third trimester (FT3). Asterisk (*) indicates that *p* is less than 0.05.

**Figure 8 life-11-00318-f008:**
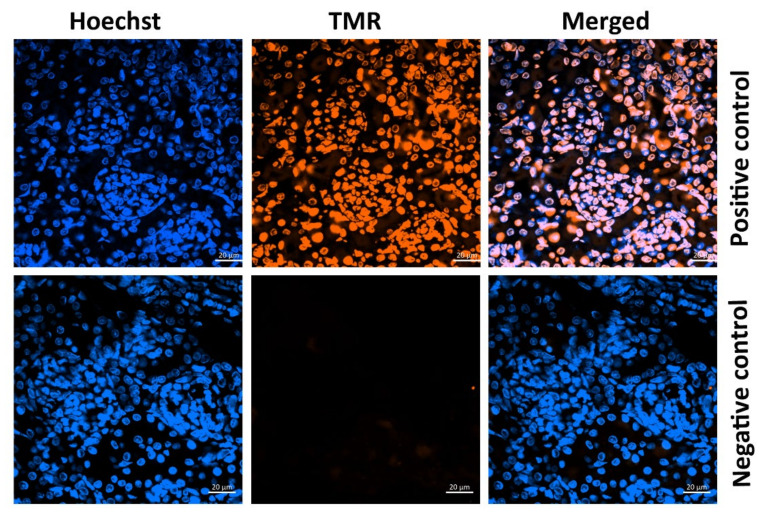
Micrographs showing the positive and negative controls stained with tunneling magnetoresistance (TMR-red). Hoechst dye (blue) was used for nuclear labeling.

**Figure 9 life-11-00318-f009:**
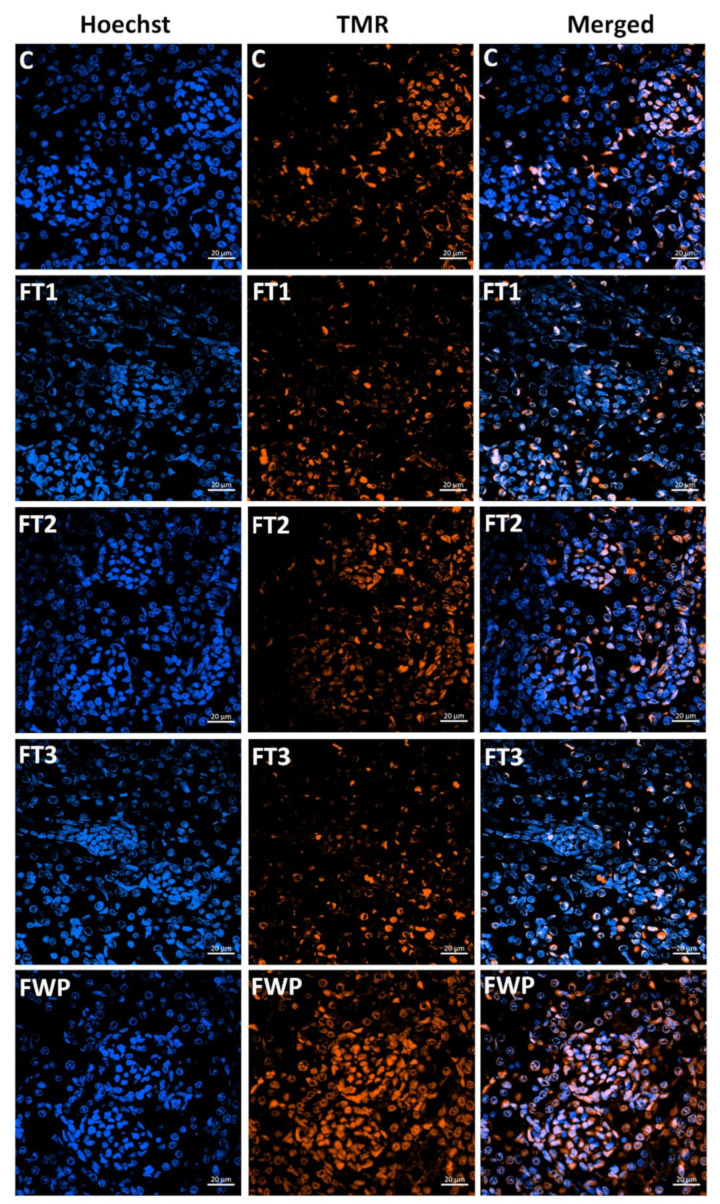
TUNEL assay staining of kidney sections from rats after fasting in the first (**FT1**), second (**FT2**), or third trimester (**FT3**), or during the whole pregnancy (**FWP**), compared to control rats (**C**).

**Figure 10 life-11-00318-f010:**
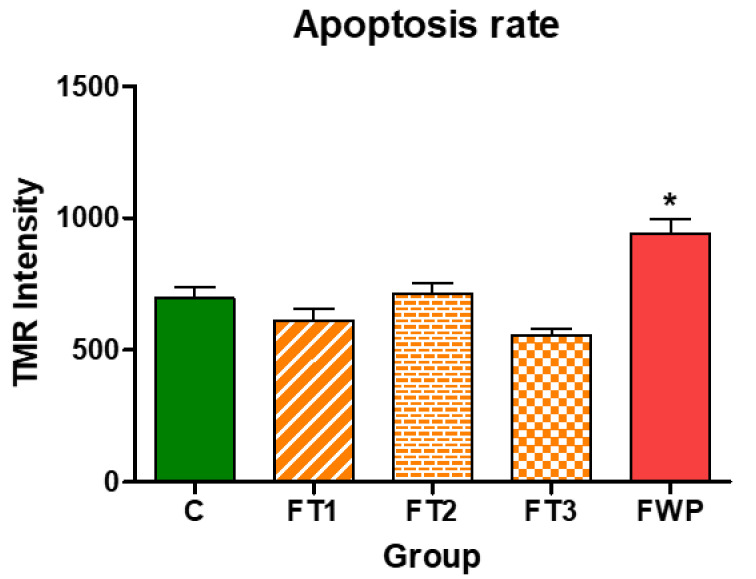
TUNEL assay showing TMR intensity in the FWP group, compared to other experimental groups. Asterisk (*) indicates that *p* is less than 0.05.

**Table 1 life-11-00318-t001:** Standard rat chow composition.

Ingredient	% Weight
Protein	20
Oil	10
Starch	42
Fibers	6
Sucrose	20
Minerals	1.5
Vitamins	0.5

**Table 2 life-11-00318-t002:** Maternal and litter measurements.

Measurements	C	FT1	F2	FT3	FWP
Female weight at mating (g)	234.2 ± 18.7	238.5 ± 24.2	240.7 ± 21.1	236.5 ± 11.4	229.3 ± 8.8
Accumulated food intake (g)	356.5 ± 25.7	345.1 ± 16.20	352.1 ± 24.2	337.6 ± 38.6	284.4 ± 26.5 *
Accumulated water intake (mL)	781.2 ± 127.4	753.6 ± 138.9	714.2 ± 136.7	789.1 ± 133.4	523.5 ± 114.1 *
Gestational weight gain (g)	84.25 ± 3.30	86.00 ± 17.66	80.50 ± 17.66	66.33 ± 12.05 *	55.50 ± 14.41 *
Gestational length (day) ^â^	21.66 ± 0.51	22.00 ± 0.00	22.00 ± 0.00	21.50 ± 0.54 *	22.00 ± 0.00
Body weight at delivery/initial weight (%)	8.82 ± 2.90	−2.18 ± 4.85 *	−3.60 ± 4.37 *	−1.68 ± 3.15 *	−9.38 ± 3.11 *¥
Litter size (pup)	9.60 ± 2.51	10.83 ± 2.32	11.33 ± 2.42	10.33 ± 1.37	9.70 ± 2.45
Pup weight (g)	6.52 ± 0.68	5.65 ± 0.59*	5.67 ± 0.64 *	5.53 ± 0.53 *	5.23 ± 0.48 *¥

^â^ Gestational length was measured to the nearest 0.5 day. C, control; FT1, exposed to fasting in the first trimester; FT2, exposed to fasting in the second trimester; FT1, exposed to fasting in the third trimester; FT1, exposed to fasting during whole pregnancy. ¥: this group significantly differs from other groups with *.

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
