# Peer review of "Long-Term but Not Short-Term Maternal Fasting Reduces Nephron Number and Alters the Glomerular Filtration Barrier in Rat Offspring"

_life, 2021, doi:10.3390/life11040318_

Round 1

Reviewer 1 Report

The authors examined the effects of maternal fasting on the fetal kidney development. It is an interesting study in an important field. My comments are as below.

  1. Several similar studies have been already published. For example, PMID: 29317010 and a thesis (Alla Alkhalefah, University of Manchester, UK).
  2. The data showing the impairment of glomerular filtration barrier is not convincing. The TEM image is not clear enough. Additionally, I suggest the authors to measure glomerular permeability, such as urinary protein or albumin, as supplement to support the findings in structure.

Author Response

Reviewer 1

The authors examined the effects of maternal fasting on the fetal kidney development. It is an interesting study in an important field. My comments are as below.

  1. Several similar studies have been already published. For example, PMID: 29317010 and a thesis (Alla Alkhalefah, University of Manchester, UK).

We thank the reviewer for his interest in our study. The first study (PMID: 29317010) was performed in our laboratory and by our group. That study was the first attempt to mimic Ramadan fasting, however, in that study we have implemented two groups: control and experimental group. In the experimental group we exposed dams for fasting during whole pregnancy period which does not represent the real fasting period in Islamic world since women do not fast during the whole pregnancy, rather they fast one month out of nine months. In the current study we implemented four experimental groups; dams exposed to fasting during whole pregnancy in order to confirm our previous findings and dams that exposed to fasting for 3 days which represent one nineth of gestation. The one nineth of fasting was in the first, second or third trimester to identify the most critical period. In addition, here we have measured glomerular size and we examined the ultrastructure of the filtration barrier which were not studied in the previous study.   

The second study, Alla Alkhalefah, was a very interesting and detailed study. However, Alkhalefah did not abstain dams from drinking water while in our study we withdrew both food and water during daily fasting period. In addition, Alkhalefah exposed dams to fasting during whole gestation while in our study we have other subgroups where dams were exposed to fasting for one nineth only. lkhalefah did not examine the ultrastructure of the filtration barrier as we did.

Therefore, we think that our study has examined new dimensions to add more values for our knowledge,

  1. The data showing the impairment of glomerular filtration barrier is not convincing. The TEM image is not clear enough. Additionally, I suggest the authors to measure glomerular permeability, such as urinary protein or albumin, as supplement to support the findings in structure.

We have changed the micrograph and hope the new one would be acceptable.

We would like to thank the reviewer for his valuable suggestion. We have now measured urinary albumin and found that FWP group has a significant increase which support the idea of altered glomerular filtration barrier. New sections regarding urinary albumin have been added to the methods, results and discussion. 

Reviewer 2 Report

The authors have done a good work portraying the effects of maternal malnutrition on fetal growth. 

Author Response

Reviewer 2

The authors have done a good work portraying the effects of maternal malnutrition on fetal growth. 

We thank the reviewer for his/her positive comment. 

Reviewer 3 Report

The authors have work hard to complete the research project which received ethics approval.

However I would recommend authors to reduce the introduction length. Please remove David Barker and instead give reference to his work.  or mention Barker et al. We do not recommend writing full names of the references.

Also if the authors can explain in the discussion why female rat off-springs were excluded from the study.

Author Response

Reviewer 3

The authors have work hard to complete the research project which received ethics approval.

We thank the reviewer for his/her positive comment. 

However, I would recommend authors to reduce the introduction length. Please remove David Barker and instead give reference to his work.  or mention Barker et al. We do not recommend writing full names of the references.

Agreed! We reduced the introduction size and we removed the name of David Barker. 

Also if the authors can explain in the discussion why female rat off-springs were excluded from the study.

The reviewer has raised an important point, and we agree that the offspring gender may influence the effect of maternal fasting. In this study, we restricted our work on male on excluded female offspring because the work was large enough for this project. We hope that the reviewer understands our limitations in term of time and money. We will look at the gender effect in upcoming studies.

Round 2

Reviewer 1 Report

No more comments.